# A Two-Stage Fuzzy Optimization Model for Urban Land Use: A Case Study of Chongzhou City

**Jinjiang Yao [1], Bingkui Qiu [2], Min Zhou [3], Aiping Deng [3,*] and Siqi Li [3]**

1   Science and Technology Innovation and Public Management Research Center of Shanghai, Fudan University, Shanghai 200433, China; jinjiangyao@126.com
2   Department of Tourism Management, Jin Zhong University, Jinzhong 033619, China; qbk@jzxy.edu.cn
3   College of Public Administration, Huazhong University of Science and Technology, Wuhan 430074, China; shijieshandian00@163.com (M.Z.); withdoca@163.com (S.L.)
*   Correspondence: DAPAipingDeng@outlook.com; Tel./Fax: +86-27-87543047

**Abstract:** Under the background of New-type Urbanization, with the continuous advancement of urbanization and the all-round development of cities, all kinds of demands are also rising. In the case of demand, it is difficult to quickly adjust from the land supply side and to guide the optimization of the structure and layout of land use is one of the methods to achieve this based on the current situation and shortage of urban land use structure and spatial arrangement. Because of the complexity, uncertainty and dynamics of the land use system, it is necessary to use an uncertain model to accurately describe and propose the approximate optimal solution, so this study analyzes the influencing mechanism of land use and optimize the land use structure under uncertainties by using a Bayesian network and fuzzy mathematical programming. Based on the results of the two stages of analysis, the cellular automata simulation is completed. The framework is applied to Chongzhou city in western China. The results indicated that the optimal land space for cultivated land is in the middle and the south based on the joint influence probability of arable land and urban construction land. The conversion probability of the area near the east is low, and the joint impact probability of construction land in all areas is generally similar except for the western protection area. After the optimization of the fuzzy planning, the optimal construction land scale is 69.42 km$^2$. Under the condition that the cultivated land's red line is guaranteed, there is still 98.87 km$^2$ of space for the increase in cultivated land. It is found through simulation that the increase in construction land would occur in the central and western parts of Chongzhou, which may be caused by the urban siphon effect. According to Monte Carlo verification, when the conversion probability exceeds 50%, the cultivated land could be turned into urban construction land, with an accuracy of 91.99%. Therefore, this proposed framework is helpful to understand the process of land use and provides a reference for making scientific and reasonable territorial spatial planning and guiding land use practice under uncertainties.

**Keywords:** two-stage land use; Bayesian network; fuzzy mathematical programming; Chongzhou city; uncertainty

## 1. Introduction

Land-use systems are complex, diverse and large, and the academic community tries to narrow the boundaries of land-use systems from the perspective of scenario setting to facilitate the use of various models and methods for research [1]. In terms of land use simulation, different land use systems, methods and models derive different driving mechanisms [2–4] containing various assumptions in the simulation process, which provide new ideas and angles for exploring the uncertainty in land use system and form our understanding of land use system today. Although there are systematic differences in scenario-based land use simulation and modeling, uncertainty does not change with the model methods and scenarios set [5], namely, changes caused by changing model

parameters and modifying a hypothesis do not change the uncertainty as well. Reducing uncertainty in land use forecasts is desirable, and some technical efforts can be made, but it is problematic to determine an "accurate" model for a specific purpose. Compared with other systems, the complexity and hugeness of land use systems do not allow repeated proofreading and collection of data that occurred in the past, which makes it more difficult to test the validity of simulation. Reducing such known uncertainties is a common problem faced by experts and scholars. What academics need to pay attention to is how to improve their understanding of the laws of land use systems and how to use these uncertainties to guide land use. It is a current problem that needs to be solved urgently, and it is also the way used to guide practice in a timely manner. Moreover, with the continuous advancement of urbanization and the all-round development of cities, all kinds of demands are also rising, so it is not feasible to reduce the demand in a short period of time [6]. Therefore, in the case of demand, it is difficult to quickly adjust from the land supply side, and to guide and play the role of land in the supply side reform elements may be one of the solutions.

With the development of land use research, more and more scholars have been inspired to capture uncertainty. The uncertainty of spatial factors will affect land use decision making, and incorporating uncertainty into the optimization of land use structure has gradually become one of the important modeling directions of the future [7,8]. Land use structure optimization and spatial optimization together provide an important theoretical basis for the formulation of territorial spatial planning. Territorial spatial planning is an important guide for land use practice, and planning often has problems with unclear planning intentions and inaccurate spatial grasp at the driving level of land use changes [9]. This is due to the inappropriate perception by the framers of uncertainties and laws in land use. In addition to emphasizing the situational and social construction in space [10], planning should also objectively describe and measure the actual existence and summarize it, including correctly understanding the law of land use and identifying various uncertainties [5]. Uncertainty is inherent. Although the tools used to study land use change may never know all the information, some uncertain methods are still expected to be applied to express and capture some of the uncertainties. Some representative studies are as follows, Verstegen et al. (2012) constructed a PCraster land use change model, which integrated the functions of visualization, simulation and uncertainty analysis. Van et al. (2012) simulated land supply with known drivers of land use, and then analyzed uncertainty using the Monte Carlo method. Meyer et al. (2012) used Bayesian network to help assess the uncertainties brought about by stakeholder knowledge modeling. Different from the uncertainty of spatial influencing factors, the uncertainty in the optimization of land use structure is often ignored. Most studies used ordinary planning to balance the land use demand of various sectors [11–15]. However, it is necessary to consider the uncertainty in the optimization of the land use structure. In order to perfect the research on this aspect, fuzzy mathematical programming is applied to it. Moreover, the uncertainty in the land use system has not been widely studied, and there are few relevant data. Some studies that take into account this problem are also always described as random distributions [16], which means that some research choses to avoid uncertainty and generally define the sources of uncertainty. The related research only analyzed the uncertainty of the influencing factors, but seldom applied the results obtained by the uncertainty method to the spatial simulation.

Mathematical models are widely used in optimization research in various fields to solve the problems of multi-constraint conditions and multi-objective optimal configuration. In the field of land use, mathematical planning has long been applied. The mathematical programming model is the optimal allocation scheme determined by mathematical programming theory [17–27]. For example, integer programming [28,29] is used to solve the optimization problem of integer variables. Dynamic programming [30] is used to solve the optimal problem of decision-making processes. Linear programming [31,32] is used to solve the one-dimensional optimal allocation problem. Nonlinear programming [33–35] is used to solve multi-dimensional allocation optimization problems. Among them, the use of integer programming needs to limit the variable to integers, so the scope of ap-

plication is not wide enough. Dynamic programming can realize the optimization of the dynamic decision-making process of linkage, but there may be a multi-dimensional solution dilemma. Generally, the time cost is high. Linear programming is simple to solve and more applicable than other kinds of programming, but only for simple linear programming problems. Fuzzy programming, random programming, chance-constrained programming, parameter programming, etc., are all linear programming methods. Stochastic programming is mainly used for stochastic processes, which can be solved by intelligent models. Parameter planning is applicable to exploring the influence of parameter changes on the final results and has a specific scope of application. Both types of planning are scenario specific. Fuzzy programming and chance-constrained programming have a relatively wide range of applications. They can express the general decision-making process and can be solved even when constraints are violated and the value of variables is not clear. The results can be used as a reference for decision makers. The applicability of nonlinear programming is not as broad as that of linear programming because of its multi-dimensional characteristics, but nonlinear programming has better performance for some complex problems.

In view of the shortcomings of the above models, the purpose of this paper is to fully consider the uncertain factors in the land use system and optimize the land use quantity and the spatial layout by combining cellular automata model so as to provide quantitative support for decision makers. Based on the complexity, uncertainty and dynamic characteristics of land use systems, the modeling method of capturing uncertainty is used to find more spatial information from the perspective of uncertainty, and the function of multiple spatial impact factors was constructed to depict the law of land use impact. To a certain extent, this expands the depth of the analysis of influencing factors, provides certain theoretical significance for the study of land use development and further deepens the cognition of the uncertainty in the process of land use. Under the background of the new urbanization strategy proposed by the Chinese government to pay more attention to the coordinated development of population, economy, society and ecological environment [36], based on the current situation and shortage of urban land use structure and spatial arrangement, combined with regional development policies, starting from the reform of land supply side structure, using the method of Bayesian network and fuzzy mathematical programming, this paper makes an in-depth analysis of the influencing mechanism of land use and the optimization of land use structure in two stages. It is helpful to provide theoretical support for the formulation of territorial space planning. The optimization result aiming at economic carbon efficiency is combined as well to provide a certain reference for the sustainable development of urban land use. China's New-type Urbanization and agriculture-oriented cities will be used as case demonstration.

## 2. Methodology

### 2.1. Two-Stage Land Use Optimization Model

Figure 1 shows a two-stage land use optimization framework under uncertain conditions. This study divides land use analysis into two stages. The first stage is to analyze land use influencing factors through a Bayesian network. It performs well in dealing with uncertainties and can reflect a more real world. The second stage is to optimize land use through fuzzy mathematical programming. In addition to economic and social constraints, this stage also involves environmental constraints marked by carbon emissions. In order to link the two stages, cellular automata are used to finally realize the analysis results of the two stages, and the neighborhood influence is added.

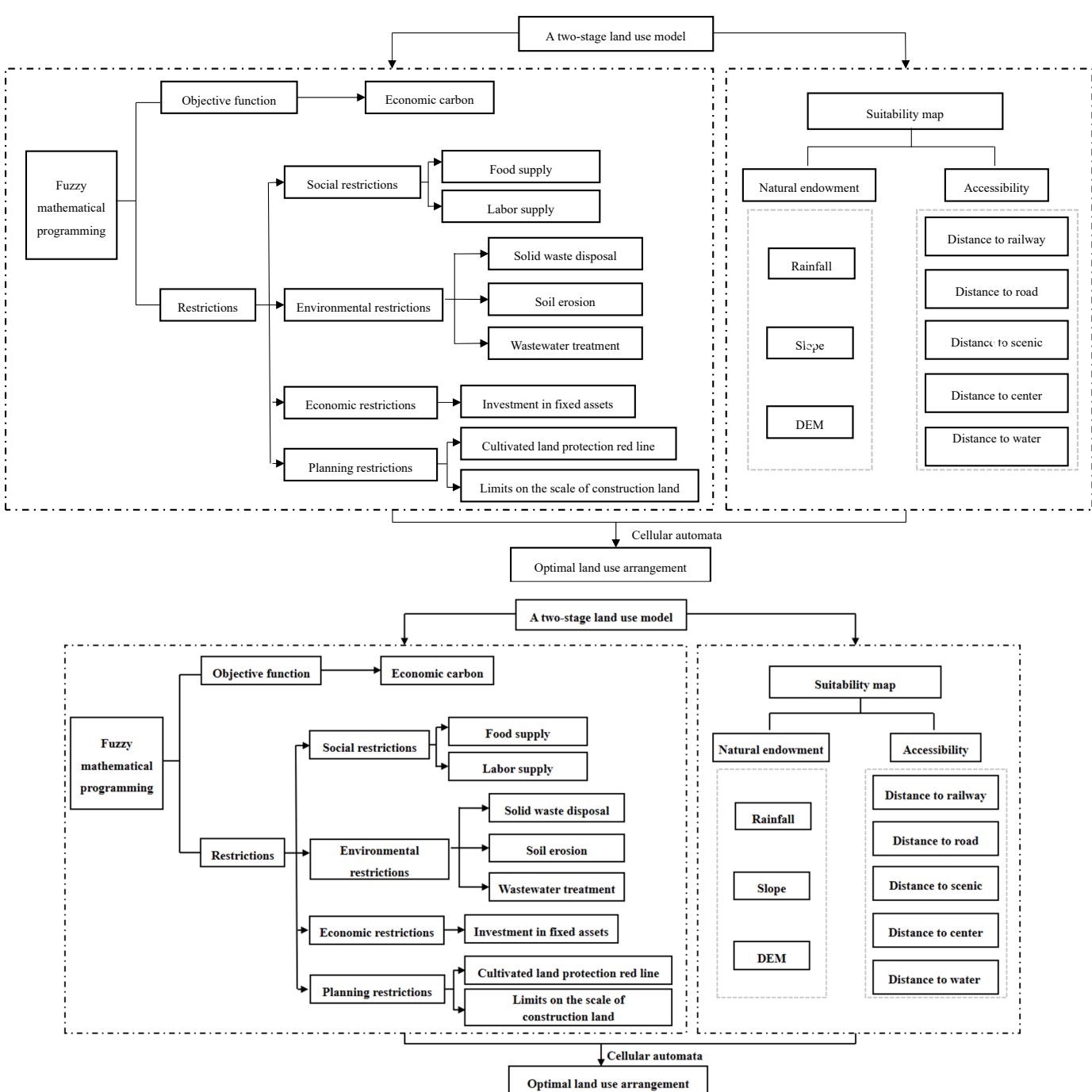

**Figure 1.** Two-stage optimization framework.

### 2.1.1. Bayesian Network for Land Use

For the occurrence or non-occurrence of land use events, there are usually only two probabilities of 1 and 0, that is, either occurrence or non-occurrence. Few studies have considered the occurrence probability of land use events, and Bayesian network has the ability to solve this problem. Bayesian network is particularly flexible, and new information can often be found during modeling to improve the performance and applicability of the network [37]. It is a way to deal with uncertainty [12] and provides more information for interpretation of the evaluation results. Bayesian network clearly shows the complex relationship between land use change, its driving factors and its change process with its good graphical description method. Although the influence of each uncertain factor on the estimation results can be measured by sensitivity analysis, it requires the estimation

of complex partial derivatives. Bayesian network not only does not need complex differentiation, but it is closer to the reduction of the real situation, in which the actual process is probabilistic in nature [38]. A node $X$ points to another node $Y$, where $X$ is the parent of $Y$. All parent nodes can be used as a set $X = \{x_1, x_2, \ldots x_k\}$, child nodes can also be represented by the set $Y = \{y_1, y_2, \ldots y_k\}$, represented as a directed acyclic graph, which is the advanced form of Markov chain. The analysis of land use suitability can also be completed by Bayesian network. The Bayesian formula is shown in Formula (1):

$$P(X|Y) = \frac{P(Y|X) \cdot P(X)}{P(Y)} \tag{1}$$

In Formula (1), $P(X|Y)$ represents the probability that the parent node $X$ is in a certain state under the condition of child node $Y$, that is, the posterior probability. $P(Y|X)$ is the same. $P(X)$ is the prior probability of $X$ without regard to $Y$, and $P(Y)$ is the same thing. According to Bayesian theory, in land use analysis, it is usually assumed that all factors are independent of each other, so their relationship can be expressed by the joint probability Formula (2):

$$P(x_1, x_2, \ldots x_k) = P(x_k|x_1, x_2, \ldots x_{k-1}) \cdots P(x_2|x_1)P(x_1) \tag{2}$$

The construction of a Bayesian network involves qualitative and quantitative steps. The first step is to formulate the node and its network structure, and the second is to calculate the parameters of each node. It is found that policy and socio-economic factors show complex and close links under the influence of natural factors such as resource endowment and spatial geographical characteristics [4]. At the same time, the design and implementation of land planning will be affected by local socio-economic factors and external processes [6]. This study extracts candidate variables from spatial data, constructs the network structure according to common sense, expert knowledge and statistical correlation between variables and avoids creating parameter sets with excessive data as much as possible [39]. Therefore, in terms of spatial factor analysis, historical state, elevation, slope, accessibility, natural endowment and other factors are taken into consideration in the construction of the Bayesian network, as shown in Figures 2 and 3. First is the linear features of the impact of land use, railway effects on the city of the population, material and information exchanges with the outside world, belonging to the clear on the larger influence on urban development tool, so it leads to the improvement of the region and all aspects of development, both in cultivated land activities and urban economic activities, and can produce a good liquidity. In addition to assuming the connection with the city, the highway also serves as a bridge for communication within the city, facilitating the effective connection and interaction of internal elements. So linear features can lead to strip development around them. Secondly, point features have a certain radiating effect on urban construction, such as scenic spots and commercial centers. These point features are active centers of population and capital in cities, which attract population and investment, are easier to form intensive land use and have more potential to lead development. Moreover, natural endowments are vital to both urban and agricultural development. For agriculture, water supply will directly have an important impact on crop quality and yield. Rice, the main food crop in Chongzhou City, will have its yield affected by the lack of water resources. At the same time, water is an important resource guarantee for urban construction. Sufficient water supply will support urban life and production and, vice versa, will limit urban sustainable development. In order to improve the quality of water resources, provide a good resource endowment for urban life and production and improve the ecological and landscape conditions of the city, Chongzhou City has implemented measures such as the 'water control ten' river (lake) chief system, black and odorous water treatment and comprehensive water environment management. Therefore, this shows that the current agricultural development and urban construction are both limited by water and dependent on water.

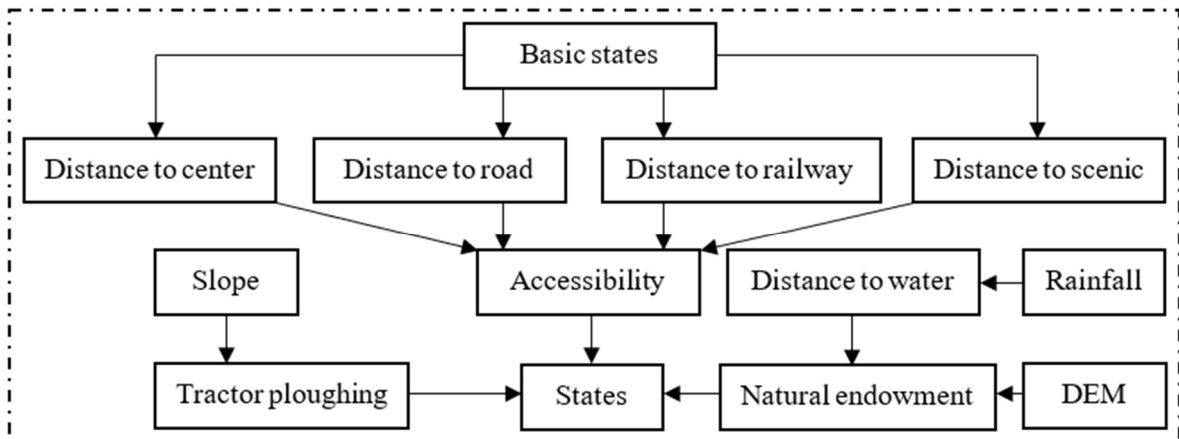

**Figure 2.** Bayesian networks for cultivated land.

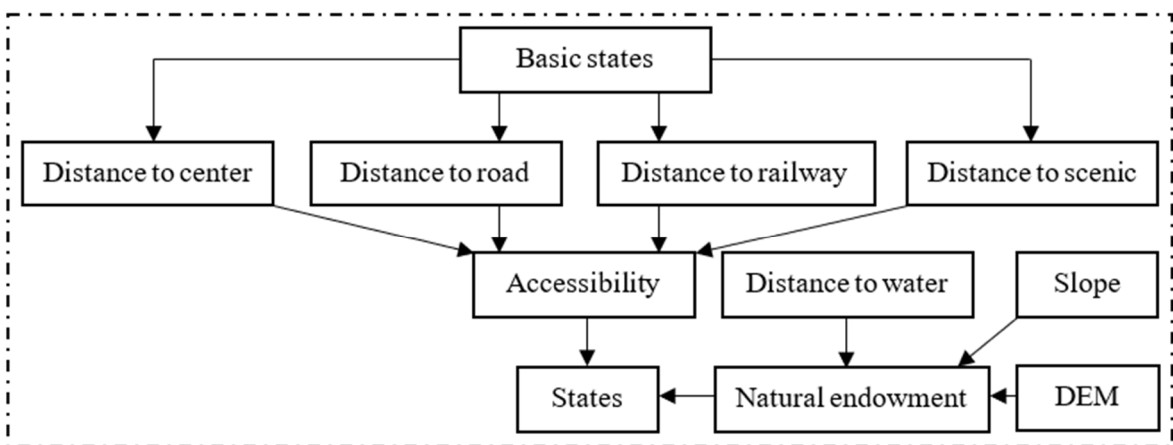

**Figure 3.** Bayesian network of urban construction land.

Bayesian network can help to search various laws in land use systems, and Gaussian radial basis functions can help to express these laws. The various influencing factors of land use change have different forces, so the attraction of each factor to different land types is also different. For example, the closer the area is to the city center, the easier it is to develop into urban construction land. The attraction of the city center is higher than that of the county center and the town center. The land close to the highway is easier to form strips of urban construction land on both sides of the highway, and the higher the grade of the highway, the easier it is. Similarly, rivers have a huge impact on urban development, and urban construction land is often gathered along the banks. Therefore, there are differences in the influence ranges of different influencing factors. It is generally believed that there is a certain rule with distance. In order to express the difference of forces generated by distance, a Gaussian radial basis function is adopted, as shown in Formula (3):

$$P_k(x_k|m) = exp\left(-\frac{1}{2\sigma_k^2}(x_k - \overline{x}_k)^2\right) \tag{3}$$

where $P_k(x_k|m)$ represents the probability of developing into a certain land type, and $\sigma$ represents the standard deviation of the influencing factor variable. The larger the value of $\sigma$, the wider the influence range of the variable, while the smaller the value, the more concentrated the influence range. $x_k$ represents the value of the variable, and $x_k$ represents the central value, that is, the variable value most likely to develop into the target land type, at which time the probability of $P_k(x_k|m)$ reaches the maximum.

### 2.1.2. Land Use Optimization Mathematical Programming Model

Next, we estimated the uncertainty in the land use system is crucial for investigating land use policies, evaluated urban risk response strategies and quantified the impact of land use change on climate and the environment. Uncertainty involves a wide range, and this study uses a small entry point to express one aspect of the uncertainty, namely, the uncertainty of the target conditions for economic carbon efficiency, as shown in Formula (4):

$$Max\ Z = \frac{\sum_{i=1}^{n} x_i \cdot UB_i}{\sum_{i=1}^{n} x_i \cdot WC_i + \sum_{i=1}^{n} x_i \cdot PC_i + \sum_{i=1}^{n} x_i \cdot EC_i + \sum_{i=1}^{n} x_i \cdot SC_i + AC} \tag{4}$$

In Formula (4), $x_i$ represents the area of each type of land; *Max* means the maximum economic benefit; and $UB_i$ represents the level of economic benefit generated per unit land area ($kg/km^2$). In this study, the gross output of primary industry, secondary industry and tertiary industry represents the gross output of cultivated land and urban construction land. $UB_i$ is the ratio of the gross output of each type of land to the area of each type of land before optimization. *Min* represents the minimization of carbon emission. $WC_i$ represents the carbon emissions ($kg/km^2$) generated by wastewater quantified on all kinds of land, which can be obtained by the ratio of the amount of cultivated land wastewater and urban wastewater to the area of all kinds of land obtained from statistical data. $PC_i$ is the carbon emission from human body on various types of land ($kg/km^2$). $EC_i$ represents the carbon emission from energy consumption of various types of land ($kg/km^2$). $SC_i$ represents the carbon emissions from solid waste generated from various types of land ($kg/km^2$). *AC* stands for carbon emissions from livestock and poultry farming. In order to more clearly understand the relationship between economic efficiency and carbon emissions, this research integrates economic benefit targets and carbon emission targets to explore the economic carbon efficiency, that is, how much economic benefit each unit of carbon emission produces and whether it is achieved between the sacrifice of the climate and the economic benefit balance. Therefore, the objective function is optimized.

(1) Environmental constraints

Since both agricultural and urban development have an impact on the soil and water environment, it is necessary to consider the status of soil. In the Formula (5), $r$ represents soil and water loss rate (%). $A_r$ represents the area of loss ($km^2$). Both farmland operation and urban production and living will produce sewage. If sewage is discharged into the environment without treatment, it will not only pollute the water environment but also cause excessive heavy metals and poisons in the soil, which will increase the burden of the ecological environment and make it unbearable, affecting the environmental sustainability. In the Formula (5), $G_i$ represents the sewage discharge coefficient ($ton/km^2$) generated on a certain type of land, which can be obtained by the ratio of sewage discharge to the area of each type of land before optimization. $P_G$ denotes sewage treatment plant capacity (tons). Both agricultural production and urban production and life need to consume a large number of solid materials and produce a large amount of solid waste. Considering that the rural solid waste is treated on-site, which is not easy to count, only the solid waste generated in the city is considered. However, the processing capacity of urban solid waste is limited, so certain constraints must be met. In Formula (5), $USW_i$ represents the discharge coefficient of solid waste generated on a certain type of land ($ton/km^2$), which can be obtained by the ratio of the amount of solid waste generated by the city to the area of urban construction land before optimization. *PSW* stands for solid waste disposal capacity (tons).

$$\sum_{i=1}^{n} r \cdot x_i \lesssim A_r$$
$$\sum_{i=1}^{n} G_i \cdot x_i \lesssim P_G \tag{5}$$
$$\sum_{i=1}^{n} G_i \cdot x_i \lesssim P_G$$

(2)   Economic constraints

The fixed asset investment required for agricultural production includes mechanical farming, electromechanical irrigation, mechanical seeding, mechanical harvesting, transportation roads and production facilities, etc. For Sichuan Province, with many mountains and hills, it belongs to an area with a high level of mechanization, so the investment ratio of fixed assets is also large. The fixed asset investment in urban construction has been the main part of the total investment. In Formula (6), $DSFAI$ represents the total investment (CNY 100 million); $SFAI_i$ represents the investment per unit land area (CNY 100 million). In this study, the fixed asset investment in primary industry, secondary industry and tertiary industry represents the fixed asset investment in cultivated land and urban construction land respectively. $ISFAI$ represents import investment (CNY 100 million). $OSFAI$ represents export investment (CNY 100 million). According to Formula (6), the per capita annual ration consumption is about 135 kg:

$$\sum_{i=1}^{n}(SFAI_i \cdot x_i) + ISFAI - OSFAI \gtrsim DSFAI \tag{6}$$

(3)   Social constraints

Agricultural products are one of its main production targets in the key county-level city for agriculture and ecological protection, so the balance of the food supply is particularly important. In Formula (7), $US_i$ represents the amount of grain produced per unit land area ($kg/km^2$), which can be obtained by the ratio of grain yield to cultivated land area before optimization. $D$ stands for grain demand (kg). Agricultural production and urban secondary and tertiary industries both need labor. In Formula (7), $UL_i$ represents the labor force per unit land area ($person/km^2$). In this study, the employees in the primary industry, the secondary industry and the tertiary industry represent the labor force on cultivated land and urban construction land, and the ratio of them to each type of land area before optimization is taken as the value of $UL_i$. $TL$ stands for total labor force (people).

$$\begin{aligned} \sum_{i=1}^{n} x_i \cdot US_i &\gtrsim D \\ \sum_{i=1}^{n}(UL_i \cdot x_i) &\lesssim TL \end{aligned} \tag{7}$$

(4)   Programming constraint

The planning conditions mainly include the restriction of cultivated land protection red line and the restriction of construction land area. In Formula (8), $FC$ represents the minimum cultivated land protected area ($km^2$). $FB$ refers to the maximum area of construction land ($km^2$) A 338 $km^2$ farmland protection red line has been set, accounting for 31.62% of the total area.

$$\begin{aligned} x_i &\geq FC, \ i = 1 \\ x_i &\leq FB, \ i = 2 \end{aligned} \tag{8}$$

(5)   Other Constraints

$$x_i \geq 0, \ i = 1, 2 \tag{9}$$

In Formula (9), $x_1$ represents cultivated land. $x_2$ represents urban construction land.

(6)   Carbon effect estimation

Before estimating the carbon effect, it is necessary to define the organizational boundary. The organizational boundary set in this study is the carbon effect within the range of cultivated land production activities and urban consumption, which mainly includes carbon sources, carbon emissions generated by mechanical farming and urban production and living energy consumption in cultivated land activities, carbon emissions from



farmland and municipal wastewater treatment, carbon emissions from municipal solid waste treatment, carbon emissions produced by the human body and carbon emissions from livestock and poultry farming. In terms of carbon sinks, cultivated land, shrub land, woodland and water are all included.

### 2.1.3. Cellular Automata Simulation

Cellular automata determines cellular state through transformation rules [40]. The improved transformation rules not only include the influence of cellular state and neighborhood action at the previous moment but also include quantified social, economic, spatial and other factors:

$$S_{t+1} = f(S_t, N, \ldots) \tag{10}$$

In Formula (10), $S$ refers to the set of all possible states, $f$ is the transition rule, $N$ is the neighborhood of the cell, $t$ moment is the current moment and $t + 1$ represents the next moment. In fact, cellular automata take into account spatial explicit weighting factors in suitability, rather than completely random allocation. Each cell's land type is considered the primary type, and that cell is masked, while all other land types are considered the "claim" type and compete for that cell. The "claim" type of cell that influences the possibility of the future land type of the principal cell must be both inherently suitable and close to the principal cell. This study adopts a neighborhood range of $3 \times 3$, with a kernel as shown in Figure 4.

**Figure 4.** Cellular automata kernel.

This study defines three basic scenarios affecting land use change and the basic settings for each scenario. The first situation is the role of neighborhood. If there is an urbanized area around, the central cell is also inclined to develop into an urbanized cell under the influence of the neighborhood state, which is manifested as the concentration of the urbanized area. In the second case, the protected area is regarded as the forbidden area for development, and it will not be developed into an urban area no matter whether it has the conditions for the development of urban area. Third case: In the case of limited city size, in order to ensure the balanced improvement of various functions of the city, it is not feasible to consider only a single goal. Therefore, the consideration of the red line of cultivated land protection and the control of construction land scale is added. This paper mainly studies two types of land with fierce competition, namely, cultivated land and urban construction land.

## 3. Case Study

Chongzhou is located in southwestern China, covering an area of 1089 km², located between 30°30′~30°53′ N and 103°07′~103°49′ E (Figure 5.). In terms of geographical conditions, the terrain of Chongzhou is high in the northwest and low in the southeast. The west is mostly high mountains, and the terrain is steep. High mountains account for 38.4% of the total area. It has mild climatic conditions, abundant agricultural conditions, more rainfall and less sunshine. In terms of history and culture, Chongzhou is one of the cradles of ancient Shu civilization and the birthplace of agricultural civilization. In terms of economy and society, the GDP of Chongzhou in 2017 was CNY 30.04 billion,

and the proportion of primary, secondary and tertiary industries was 12.1:49.5:38.4. The permanent resident population is 664,800, and the number of people employed in the primary, secondary and tertiary industries is 110,500, 282,800 and 149,600, respectively. Chongzhou is located in the west of Chengdu City, in the middle and upper reaches of Minjiang River, and is an important water source and ecological conservation area of Chengdu Plain. According to the strategy of "advancing to the east, expanding to the south, controlling to the west, transforming to the north, and providing excellent" in the General Plan of Chengdu City (2016–2035), Chongzhou is mainly responsible for the development of green industries and providing ecological services. Meanwhile, it strictly implements the ecological red line control, with cultivated land and woodland occupying the main position. According to existing data, generally speaking, cultivated land and urban construction land are mainly distributed in the central and eastern regions. Agriculture and urban construction will form a competitive relationship in terms of terrain. Thus, elevation and slope both constitute the main influencing factors of agricultural development and urban construction. In addition to the above linear features, namely, point features, elevation and slope conditions, for agricultural development, there are also natural conditions such as rainfall, soil productivity and heat that have a certain impact on the characteristics and productivity of agricultural products in the region. Taking into account the accessibility of data, this study selects rainfall as a representative to participate in the construction of the Bayesian network. In order to test the performance of the model, the Bayesian network modeling was replicated in Chongzhou ecological service zones, agricultural protection zones and major urban areas within Chengdu. On the one hand, natural endowment drives the stepwise development of land use. On the other hand, as one of the key pilot cities of New-type Urbanization in western China, it is still in the stage of urban expansion with highly developed social and economic activities, so it is suitable for this model. The general situation of Chongzhou is shown in Figure 6.

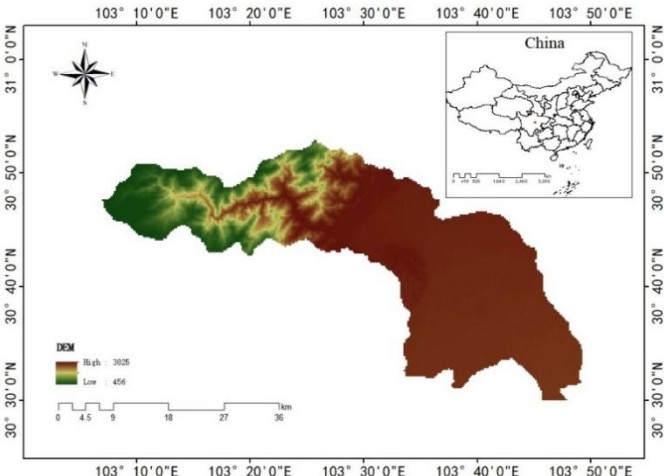

**Figure 5.** A location map of Chongzhou.

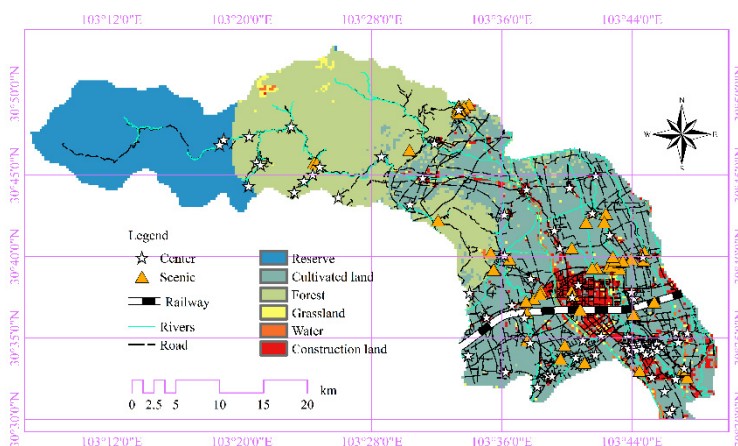

**Figure 6.** Overview of the study area.

The data mainly came from Chengdu Yearbook 2018, Chongzhou Yearbook 2018 and Sichuan Statistical Yearbook 2018, which recorded the economic and social data in 2017. The elevation and spatial land use data are from the geographical science and resources institute of the Chinese Academy of Sciences and Tsinghua University; China's present situation of land use remote sensing monitoring database 30 m global land use data (http://data.ess.tsinghua.edu.cn/fromglc2017v1.html accessed on 6 March 2019) and spatial rainfall data were collected from National Tibetan Plateau Scientific Data Center [41,42]; and spatial data such as roads, water systems, attractions and commercial centers are derived from OpenStreetMap database. According to China's industrial structure and the status quo of land use, combined with the classification standard of China's land use status in 2017, this study divides land types into cultivated land, woodland, shrubby land, water area, protected area and urban construction land. The data on livestock and poultry rearing quantities, feeding cycles, electricity use, fuel use and management costs are from Chongzhou Yearbook 2018, Sichuan Statistical Yearbook 2018, Sichuan Survey Yearbook 2018, National Cost-Benefit Compilation of Agricultural Products 2018 and statistical data of Ministry of Agriculture and Rural Affairs, PRC (http://zdscxx.moa.gov.cn:8080/nyb/pc/search.jsp accessed on 14 December 2021). Nitrogen emissions, $CO_2$, $CH_4$ and $N_2O$ emissions factor from the 2006 national greenhouse gas inventory guide, the provincial greenhouse gas inventory preparation guide (2011) and the United Nations Food and Agriculture Organization (http://www.fao.org/faostat/zh/#data/GM accessed on 16 July 2021). Data on the $CO_2$-eq emission factors of standard coal are from China Energy Statistical Yearbook 2018, and the $CO_2$-eq emission factor of electric power data are from literature. [43,44]. To avoid double counting, the organization boundaries of this study are beef cattle, pigs, mutton sheep and meat poultry, as well as the production area, feed area and management area of breeding in Chongzhou [45,46], as shown in Figure 7.

The classification of land use, the sign of land use conversion and the carbon emission form of the study are explained as follows: (1) Biomass storage and carbon sequestration of agricultural products are not considered in this study. (2) Because of the forest and herbs, there are many different crops and the waters of the biomass, and this paper summarizes land use cover change by the change of land type. The vegetation on the land's net ecosystem productivity (NEP) is also changed, and combined with the geographical position and climate characteristics, they determine the net ecosystem productivity of different land types. The average carbon sequestration capacity of different land types is also calculated. (3) The carbon emission form studied in this paper is $CO_2$, that is, $CO_2$ generated by various land use activities.

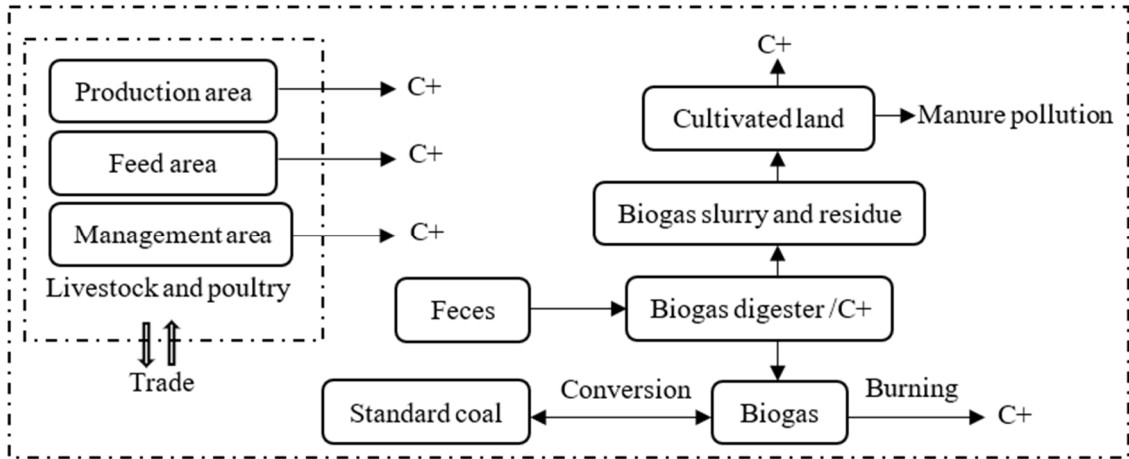

**Figure 7.** Organizational boundaries of this study.

## 4. Results and Discussion

### 4.1. Influencing Factors of Land Use: Bayesian Network

A Bayesian network is constructed and completed in Netica software (www.norsys.com, accessed on 16 July 2021), using data from railways, highways, scenic spots, commercial centers, slopes, water area, rainfall, elevation and other data processed by ArcGIS. Considering that the area of the research area is the size of a county, this research created 500 random sample points to train the Bayesian network with the sample data. Since the creation of random points results in the absence of values in the attributes of some points, 4 points were eliminated, and the total sample size became 496. The coordinate system of all kinds of spatial data was unified as GCS_WGS_1984, and all data processing was carried out in this coordinate system. The vector spatial data were converted to raster data through the element-to-raster operation, and the pixel size was 250 × 250 m. By generating the nearest neighbor table and extracting the point of arrival, all kinds of spatial attributes were linked with land use, and the attribute characteristics of each grid cell were obtained. Statistical spatial attribute data were the input type of the Netica software. Modules are constructed in Netica, and each module was divided into 10 continuous interval states. The initial probability of each module is equal so as to conduct initial exploration, determine parent nodes and child nodes according to the model design and form a directed acyclic graph.

Figure 8 shows the probability map of influencing factors of cultivated land, and Figure 8 shows the probability map of influencing factors of urban construction land. DEM is the height, Slope is the slope, Accessibility is the accessibility to all kinds of lines and points, and so on. A state refers to protected areas, B state refers to cultivated land, C state refers to woodland, D state represents shrub land, E state represents water area and F state represents urban construction land. Among them, the data of accessibility, natural endowment and farming are only statistical data without spatial data, so the preliminary determination is based on the expert knowledge, so that the follow-up research can be carried out smoothly.

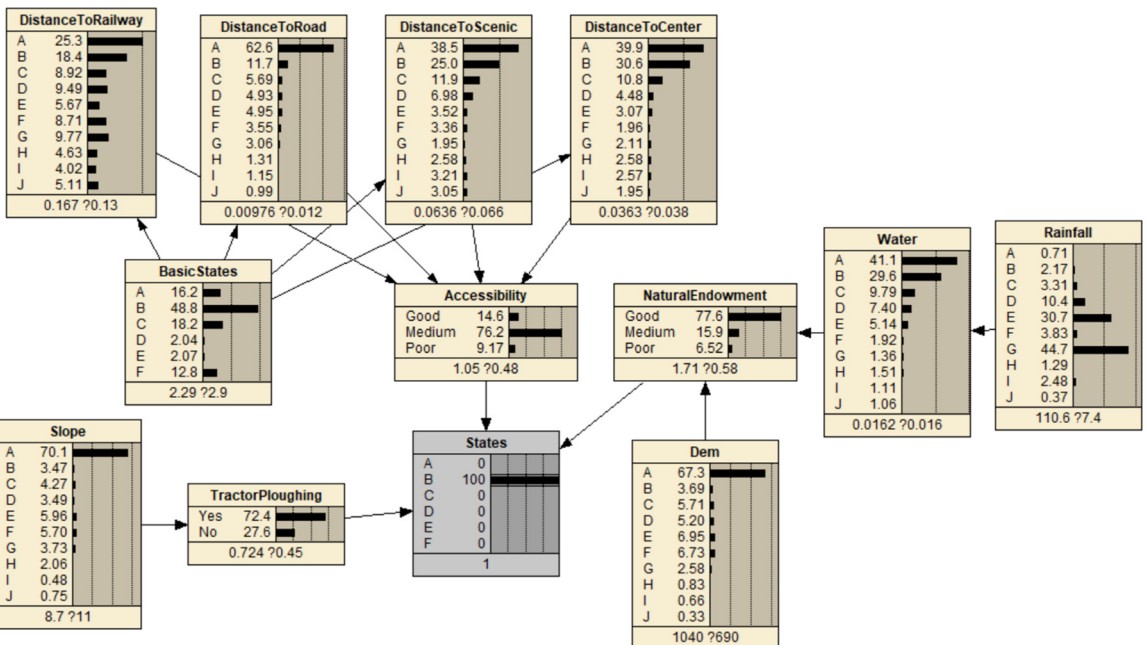

**Figure 8.** Probability diagram of factors affecting cultivated land: "DistanceToRailway" means the distance to the railway, "DistanceToRoad" means the distance to the road, "DistanceToScenic" means the distance to the scenic spot, "DistanceToCenter" means the distance to the commercial center, "NaturalEndowment" refers to natural abilities or qualities, "Tractorploughing" refers to mechanized farming or not, "BasicStates" refers to initial land use status and "States" refers to simulated land use status.

According to the results of the Bayesian network in Figure 8, the linear ground features have the same characteristics for cultivated land and urban construction land, that is, the closer the land is to railways, highways, scenic spots and commercial centers, the easier it is to develop into cultivated land, and the trend of decline is from near to far. Due to the large amount of data, all the data of the 496 sample points were not included in this paper, so the sample data were explained in words. The first is the influence brought by the railway. As an important means of transportation to other places, the railway has a wider scope of influence than other linear ground objects. In Figure 8, the states of the "Distancetorailway" node are divided into 10, each state represents a continuous interval, and the value from A to J is increasing. The value of the sample can correspond to one of the intervals, so as to achieve the purpose of grading. The same applies to other state nodes. The distance to the railway has an interval length of 4740 m, as shown in Figure 8, and the state of intervals A and B, about 9 km within range, has the greatest attraction of arable land. With the increase in distance, the influence is more and more weak. The domestic railway's furthest distance influence is also not reduced to 0, which shows that the influence of the railway is wide, but the impact is only large in a certain neighborhood. The distance to the highway is 570 m, the distance to the scenic spot is 2690 m and the distance to the commercial center is 1710 m. The reason why the length of the interval is different is because of the limitation of computing speed and memory. As can be seen from the figure, the influence rules of the three are roughly the same. The optimal impact distance of highway is interval A, about 500 m; the best impact distance of the scenic spot is between A and B, which is about 5 km. The optimal impact distance of the commercial center is between A and B, which is about 3.5 km. In terms of natural endowment, in view of the accessibility of data, elevation, slope and rainfall are selected as state nodes. Among them, rainfall can be seen as a general trend, and regions with greater rainfall have better conditions for agricultural development. Compared with the data integrity of other modules, the anomaly in interval F in the figure may be due to insufficient rainfall data sampling. The probability of elevation's influence on land use state transition presents a "cliff" distribution, that is, more than 67.3% of the cultivated land is distributed below 800 m, which indicates that greater elevation will have

a great limitation on the current agricultural development in Chongzhou. The slope is the same. Moreover, 70.1% of the cultivated land is below 5°, and more than 5° will have great restrictions on mechanized farming. The slope is divided into five grades according to the relevant provisions of topographic slope classification technology in the second national land survey: I (0°~2), II (2°~6), III (6~15), IV (15~25) and V (>25), and above 25 degrees will forbid land reclamation. In fact, the cultivated land in Chongzhou is below the degree, so the slope has a great limitation on agricultural development, which is a factor that must be taken into account in the study. The influence of water sources is similar to that of roads, and it has a greater influence on cultivated land within the range of about 1.7 km between A and B and decreases with the increase in distance. In terms of land use evaluation, mechanized tillage does not play a decisive role in land state. Areas with good natural endowment are more likely to be converted to arable land.

As can be seen from Figure 9, 59.5% of the construction land elevation is mainly kept below 800 m, and 57.3% of the construction land slope is kept below 5°. The optimal impact distance of the water area is within the range of about 1.7 km between A and B. In the two state intervals of A and B, within about 9 km, the railway has the greatest attraction to urban construction land. The optimal impact distance of highway is interval A, about 500 m; the best impact distance of the scenic spot is between A and B, which is about 5 km. The optimal impact distance of the commercial center is between A and B, which is about 3.5 km.

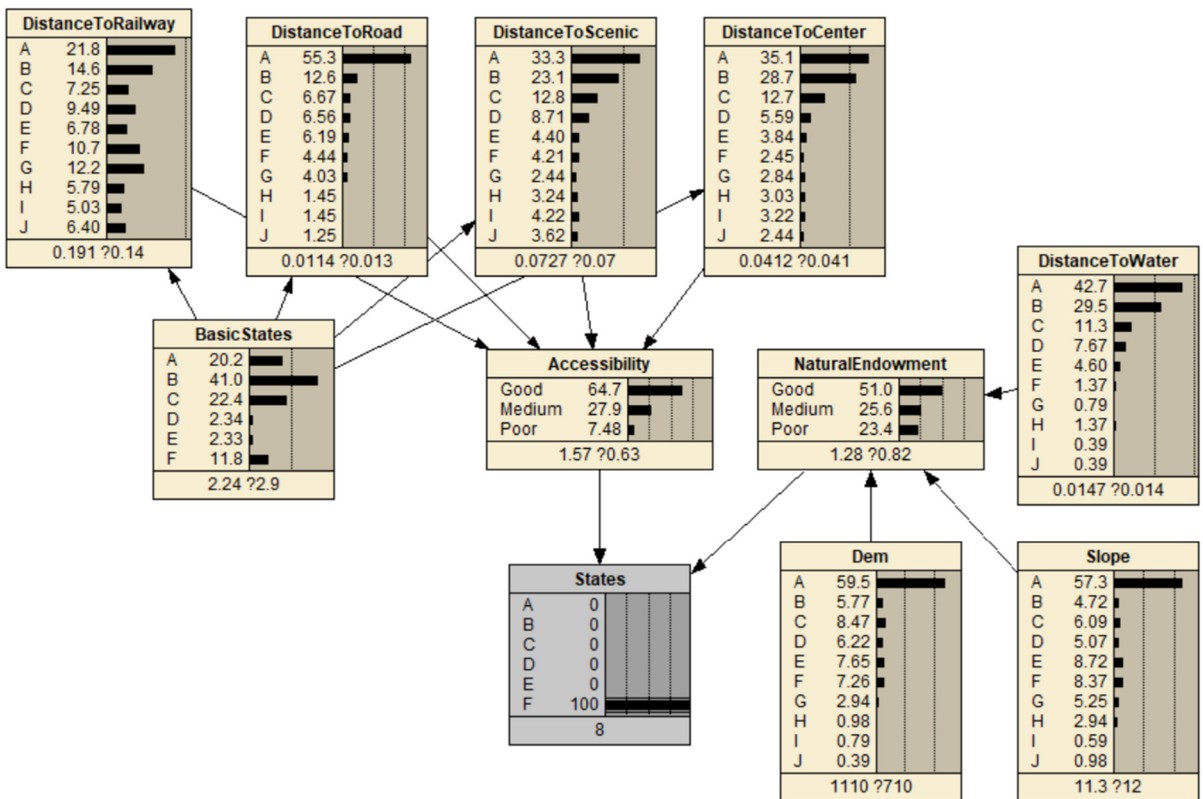

**Figure 9.** Probability diagram of influencing factors of urban construction land: "DistanceToRailway" means the distance to the railway, "DistanceToRoad" means the distance to the road, "DistanceToScenic" means the distance to the scenic spot, "DistanceToCenter" means the distance to the commercial center, "NaturalEndowment" refers to natural abilities or qualities, "Tractorploughing" refers to mechanized farming or not, "BasicStates" refers to initial land use status and "States" refers to simulated land use status.

*4.2. Comparative Analysis of Influence Factors*

It is generally believed that all influencing factors, except elevation, slope and rainfall, should present the distribution as shown in Figure 10a if their influences on land use are equidistant attenuations. However, this is not the case in fact. In this study, the influence of various factors on land use under uncertain conditions was obtained through a Bayesian network, and different results were obtained, as shown in Figure 10b. Firstly, the influence of various factors on land use is not inversely proportional to the distance. Secondly, the influence rules of various factors are not consistent. For example, the attenuation rate of railway influence is slow, while that of highway influence is faster. In order to express the influence law of each factor on land use, Gaussian radial basis function is used to express the non-inversely proportional and different influences. According to the Bayesian network constructed in Netica, the mean value and standard deviation of the influence probability of each factor can be obtained to determine the Gaussian radial basis function to express the influence probability of each factor. Figure 11 shows the Gaussian radial basis function curve of the influence factors of cultivated land drawn in MATLAB R2019a. Figure 12 shows the Gaussian radial basis function curve of influencing factors of urban construction land. Figures 11 and 12a from left to right in turn show: road (green), water (magenta), business center (yellow) and attractions (red) and railway (blue). It can be seen that under the same coordinate measurement, the influence of railway is one of the widest, the highway has the narrowest scope of influence but rail impacts few and far between. Although the road influence area is narrow, the road network is complex, and its influence cannot be underestimated. Second, the influence of the water range is relatively narrow, and the publicizing states of drainage systems are mainly the Golden Horse river, Minjiang river is flow and Black Rock stream (into Dujiangyan). The condition (within the territory of the longest) and the size of the tributary of about 180. Figure 11b shows rainfall (magenta, in millimeters), Figure 11c shows slope (red, in degrees) and Figure 11d shows elevation (blue, in meters). Slope (red, in degrees) is shown in Figure 12b and elevation (blue, in meters) is shown in Figure 12c. The analysis is described in Section 4.1.

*4.3. Affect Probability Distribution*

According to the characteristics of the Bayesian network, the probabilities of each node are independent from each other, so the joint influence probability of each factor can be calculated, which is used to express the relationship between each factor and is conducive to the transformation of independent influence probability to comprehensive influence probability. Figure 13 shows the spatial distribution of the joint influence probability of cultivated land. Figure 14 shows the spatial distribution of probability of joint impact of urban construction land. As you can see, in terms of the probability rule, arable land and urban construction land are in competition, but from the point of spatial distribution, cultivated land is the best space in central and southern areas, and the near eastern area transition probability is low and water systems are mainly distributed in the central, southern and western states. The water supply in the east is not only relatively weak, but it is also located in the direction leading to the center of Chengdu and a cluster of industrial parks. The suitability probability of urban construction land is relatively smooth, except for the low probability of the western protection area (brown-red part), the other parts of the suitability probability have little difference. The low suitability probability of a small piece of cultivated land and urban construction land in the south is caused by water area factors. Compared with other areas, this area is far from water area, so the overall probability will be slightly affected.

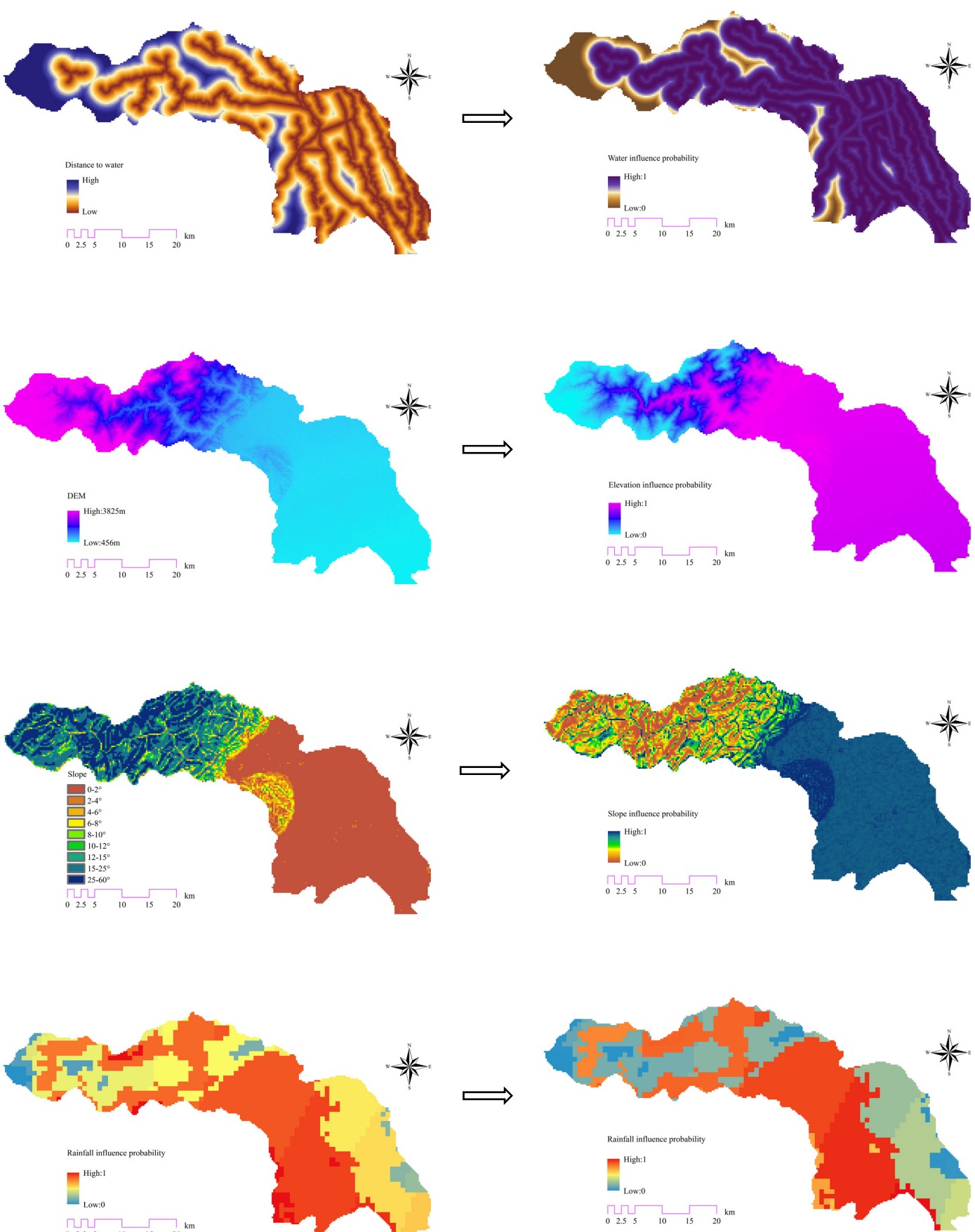

**Figure 10.** Spatial expression of various influencing factors (**a**) and influence probability distribution (**b**).

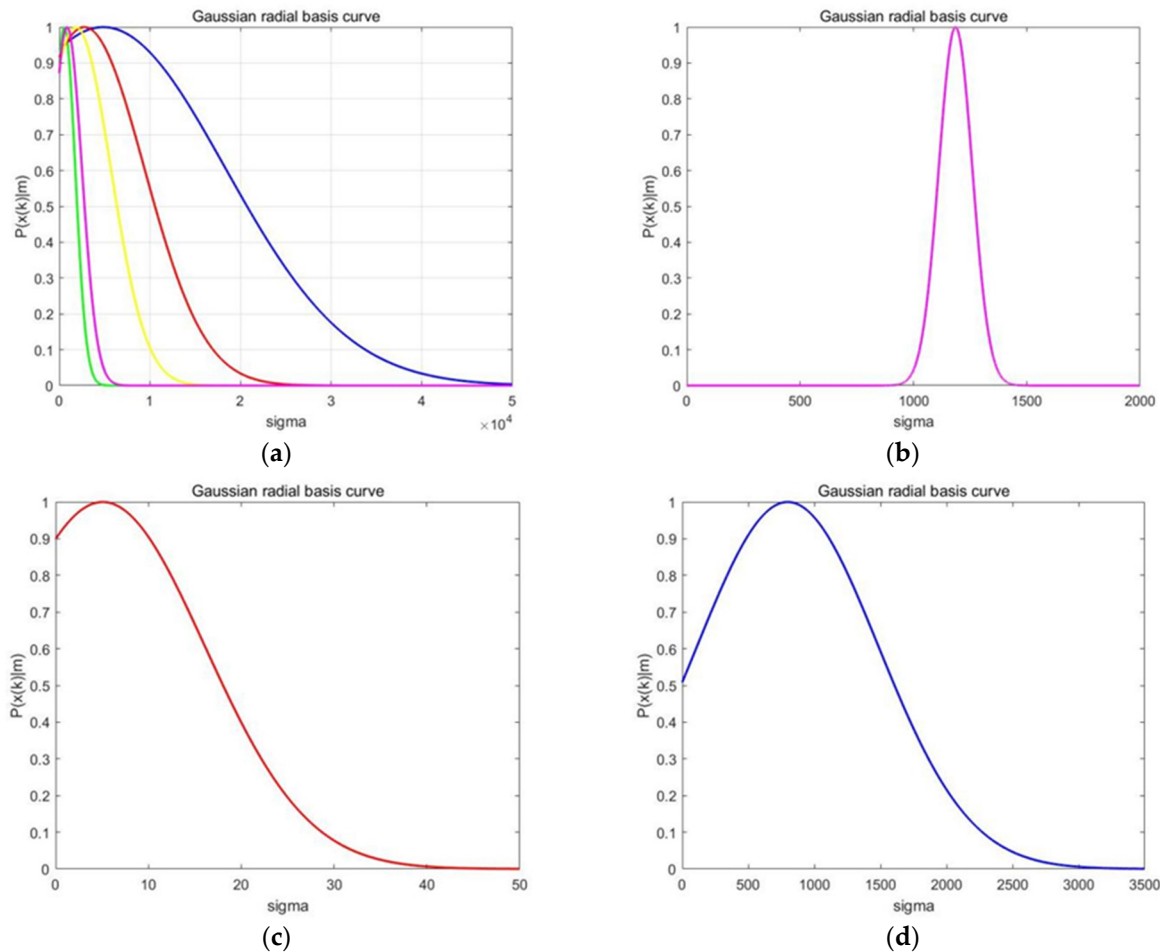

**Figure 11.** (**a–d**) Gaussian radial basis function curves of influencing factors of cultivated land.

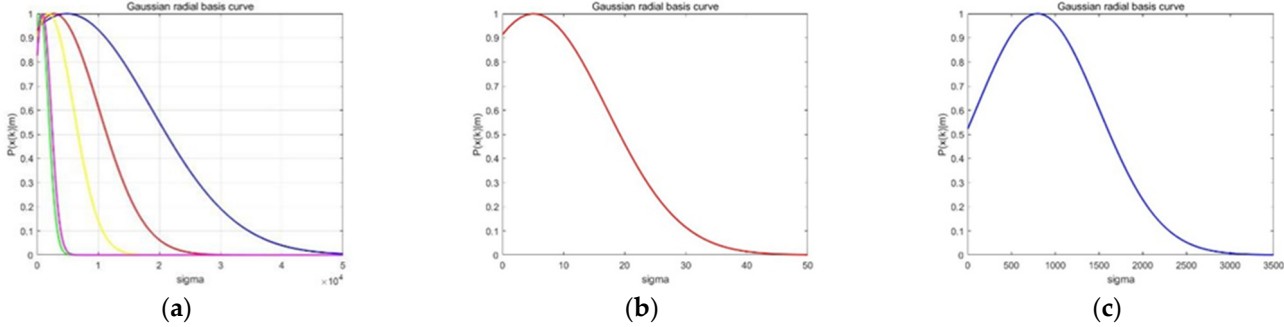

**Figure 12.** (**a–c**) Gaussian radial basis function curves of influencing factors of urban construction land.

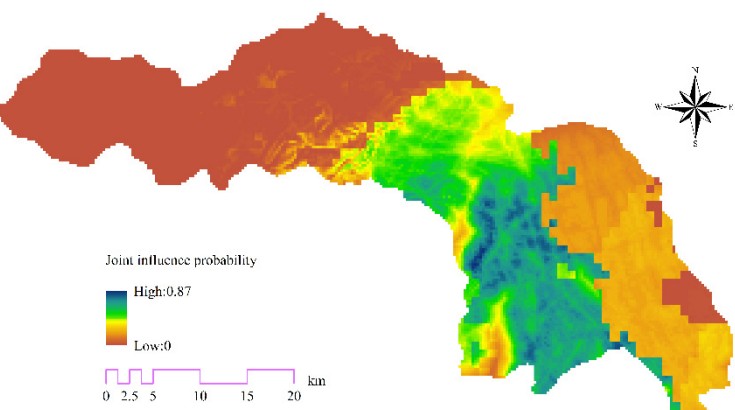

**Figure 13.** Probability distribution of joint influence of cultivated land.

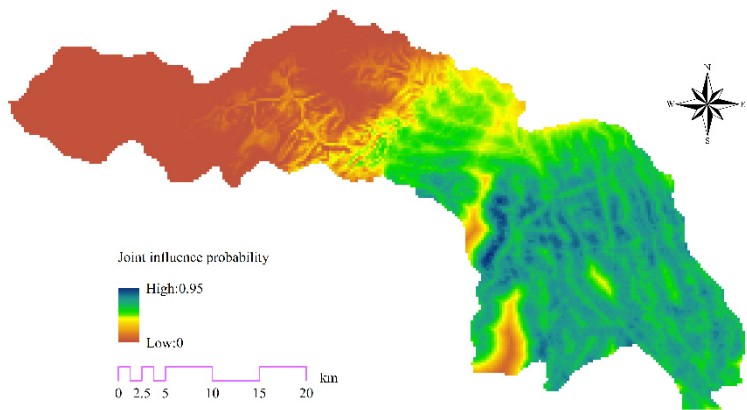

**Figure 14.** Probability distribution of joint influence of urban construction land.

### 4.4. Optimization Analysis of Land Use Structure

To solve the fuzzy mathematical programming problem in MATLAB R2019a, it is necessary to turn the fuzzy mathematical programming problem into the general programming problem. The parameters of constraint conditions include three types, including environmental parameters, economic parameters, social parameters, planning parameters and parameters of objective function, as shown in Tables 1 and 2. Finally, the value of $\lambda$ is $2 \times 10^{-8}$; $x_1$ = 436.87 km$^2$, that is, the area of cultivated land is 436.87 km$^2$; $x_2$ = 69.42 km$^2$, that is, the area of construction land is 69.42 km$^2$; $z$ = 98.38 CNY/kg, that is, the economic benefit brought by 1 kg carbon emission is CNY 98.38. The original construction land scale of Chongzhou was 68.5 km$^2$, while the construction land scale is controlled to 70 km$^2$, and the optimized result is 69.42 km$^2$. Therefore, it can be seen from the solution results that the construction land scale of Chongzhou in 2017 is within a reasonable range and can be increased appropriately but should not be too much. The cultivated land protection red line is 338 km$^2$, and the optimized results show that Chongzhou can have 436.87 km$^2$ of cultivated land. It is suggested that in addition to the reclamation of wasteland, greater efforts can be made in land consolidation and reclamation to make full use of high-quality agricultural resources. The cultivated land protection red line is 338 km$^2$. According to the calculation, the economic benefit of 1 kg carbon emission in Chongzhou is CNY 98.38, which belongs to the high economic carbon efficiency. In addition, Chongzhou is not a heavy industrial city with key carbon emission departments as its pillar industry. Its agriculture and tertiary industries are developed, and the secondary industry is dominated by green and intelligent industries. The annual carbon sink of Chongzhou is $2.3 \times 10^9$ kg, with a strong natural carbon neutralization field, which to a certain extent ensures that the carbon emissions generated in Chongzhou will not cause climate pressure in Wenchuan, Dujiangyan, Dayi and other surrounding areas and can be neutralized by itself.

**Table 1.** Constraint parameters.

| Constraint Type | Constraint Parameters | Unit | Value |
|---|---|---|---|
| Environmental parameters | Soil erosion rate | % | 2 |
| | Erodible area | km$^2$ | 8 |
| | Cultivated wastewater per unit area | kg/km$^2$ | $6.48 \times 10^6$ |
| | Municipal wastewater per unit area | kg/km$^2$ | $18.91 \times 10^6$ |
| | Total capacity of wastewater treatment | kg/year | $[1 \times 10^{10}, 1.2 \times 10^{10}]$ |
| | Solid waste per unit area | kg/km$^2$ | $58.39 \times 10^3$ |
| | Total capacity of solid waste treatment | kg | $[7 \times 10^7, 7.3 \times 10^7]$ |
| Economic parameters | Unit farmland fixed assets investment | yuan/km$^2$ | $4.16 \times 10^6$ |
| | Unit urban fixed assets investment | yuan/km$^2$ | $4.01 \times 10^8$ |
| | Import investment | yuan | $6.84 \times 10^8$ |
| | Export investment | yuan | 0 |
| | Total investment | yuan | $[2.93 \times 10^{10}, 3 \times 10^{10}]$ |
| Social parameters | Unit grain output | kg/km$^2$ | $2.51 \times 10^5$ |
| | Total grain demand | kg | $[8.97 \times 10^7, 9 \times 10^7]$ |
| | Unit cultivated labor | people/km$^2$ | 251 |
| | Unit urban labor force | people/km$^2$ | 6 312 |
| | Total labor force | people | $[5.42 \times 10^5, 6.65 \times 10^5]$ |
| Planning parameters | Cultivated land protection red line | km$^2$ | 338 |
| | Limits on the scale of construction land | km$^2$ | 70 |
| | Non-negative restriction | km$^2$ | 0 |

**Table 2.** Objective function parameters.

| Objection Parameters | Unit | Value |
|---|---|---|
| Economic benefits per unit of land area | yuan/km$^2$ | $8.60 \times 10^6$ |
| | yuan/km$^2$ | $4.41 \times 10^8$ |
| Carbon emissions from farmland wastewater treatment | kg/km$^2$ | $6.80 \times 10^3$ |
| Carbon emissions from municipal wastewater treatment | kg/km$^2$ | $19.85 \times 10^3$ |
| Carbon emissions from solid waste disposal | kg/km$^2$ | $62.28 \times 10^3$ |
| Carbon emissions from livestock and poultry production | kg | $9.59 \times 10^7$ |
| Carbon emissions from people ploughing land | kg/km$^2$ | $19.85 \times 10^3$ |
| Carbon emissions from urban construction land | kg/km$^2$ | $498.68 \times 10^3$ |
| Carbon emissions from energy use on cultivated land | kg/km$^2$ | $1.07 \times 10^3$ |
| Carbon emissions from urban energy use | kg/km$^2$ | $2.89 \times 10^6$ |

Finally, considering the application of two stage results, based on the parameters obtained from the Bayesian networks and fuzzy mathematical programming, we build the cellular automata (CA) by considering the impact of land use change of three kinds of basic situations, that is, the role of the neighborhood, the areas of conservation should be prevented from developing into urban areas as well as planning restrictions, as mentioned in Section 2.1.3. The spatial simulation of the optimized results is obtained. As shown in Figure 15, the gray area represents the protected areas and water areas that are prohibited from development and utilization, the green area represents the cultivated land, and the red area represents the construction land. By comparing the land use changes before and after simulation, it can be clearly seen that the construction land increment circled in yellow is located in the central and western parts of Chongzhou, as shown in Figure 15a. Correspondingly, the amount of cultivated land in this region will decrease, but the total area of cultivated land has space to increase. From the point of urban agglomeration, the region in the radiation zone between the state and the Chengdu core are more likely to develop into urban construction land, but the zone influenced by publicizing internal stated may also be influenced by external conditions, such as the urban siphon effect. Small cities such as Chongzhou are more susceptible to the siphon effect of large cities such as Chengdu, which is reflected in space.

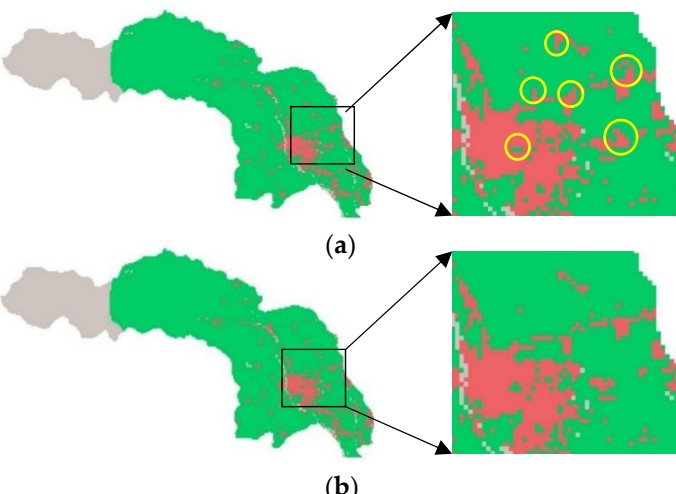

(**a**)

(**b**)

**Figure 15.** (**a**,**b**) Simulation results of land use spatial optimization of Chongzhou in 2017.

*4.5. Behavior Rule Validation*

In the 1940s, the Monte Carlo method was proposed. Monte Carlo is an experimental probabilistic method, which is based on a large number of randomized trials to describe the complex uncertainty problems encountered in practice [47]. It is a method that can be systematically and comprehensively explored, which is superior to the parameters of isolation, effectiveness analysis and single-class models in performance, especially in the ability to show the coupling error caused by coupling through the Monte Carlo test. Therefore, this study uses Monte Carlo thought to verify the transformation rules and completes simulation and calculation with ArcGIS and Excel. The Monte Carlo method determines the transition state of land use for each unit based on probability, is a more scientific method to deal with uncertain factors of land use. Considering the size of the study area, this study sampled 100,000 random points in the urban area, which can basically cover the whole area closely. Figure 16 shows the distribution of random points in a part of the area. In addition to the reserve, other non-urban cells into urban construction land cells have certain regularity and transformation rules. In order to verify the rationality of the conversion rules obtained in this study, according to the Monte Carlo idea, when the number of random points is large enough, the situation close to the actual situation can be calculated, that is, when the joint influence probability reaches that value, the cell transformation will occur. The verification results show that if policy restrictions are not taken into account, when the probability of the joint impact formulated in this study exceeds 50%, the state of cell can be changed under the condition of natural development and converted into a cell of urban construction land with an accuracy of 91.99%.

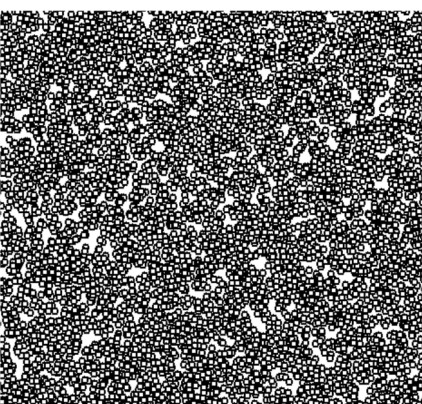

**Figure 16.** Schematic diagram of Monte Carlo random sampling.

## 5. Discussion

Based on the current situation and shortage of urban land use structure and spatial arrangement, this study analyzes the influencing mechanism of land use and optimize the land use structure under uncertainties by using a Bayesian network and fuzzy mathematical programming. Based on the results of the two stages of analysis, the cellular automata simulation is completed. The results indicated that, according to the joint influence probability of arable land and urban construction land, the best land space for cultivated land is in the middle and the south, while the conversion probability of the area near the east is low. Except for the western protection area, the joint impact probability of construction land in the other parts is generally similar. From the perspective of structure, the construction land increment of 1.5 km$^2$ can achieve the control of the construction land scale. After the optimization of fuzzy planning, the optimal construction land scale is 69.42 km$^2$. Under the condition that the cultivated land red line is guaranteed, there is still 98.87 km$^2$ of space for the increase in cultivated land. In order to make full use of high-quality agricultural resources conditions, the obtained parameters were applied to the simulation, which appeared that the increase in construction land would occur in the central and western part of Chongzhou, which may be caused by the urban siphon effect. According to Monte Carlo verification, when the conversion probability exceeds 50%, the cultivated land could be turned into urban construction land, with an accuracy of 91.99%. Therefore, this proposed framework is helpful to understand the process of land use and provides a reference for making scientific and reasonable territorial spatial planning and guiding land use practice under uncertainties.

Under conventional constraints, this study uses a small entry point to express one aspect of the uncertainty, namely, the uncertainty of the target conditions for economic carbon efficiency. The data were compared with Van et al.'s (2012) simulated land supply with known drivers of land use, and then the uncertainty was analyzed using the Monte Carlo method. Meyer et al. (2012) used a Bayesian network to help assess the uncertainties brought about by stakeholder knowledge modeling. This study considers the uncertainty in the optimization of land use structure and applied the results obtained by the uncertainty method to the spatial simulation.

As shown in the above results, the two-stage land use analysis framework constructed in this study has a good optimization effect. According to the analysis of spatial influencing factors, in the spatial pattern, cultivated land and urban construction land constitute a competitive relationship, and cultivated land is at a competitive disadvantage. If the construction land is not controlled, the agricultural space is bound to cause pressure. From the perspective of land use structure, Chongzhou can realize the growth of construction land and cultivated land area under this optimization framework, but the spatial distribution may change dynamically. Taking construction land as an example, this study simulated its spatial distribution, and the results shows that the increase could occur in the central and western part of Chongzhou, which may be caused by the radiation force of various conditions inside Chongzhou and the radiation force of the external core area of Chengdu. Although the development of the protected area is restricted, it does not constitute the conditions for the development of the urban area. Based on the analysis results of land use in Chongzhou, it can be seen that the spatial influence factors can objectively reflect the law of land use in Chongzhou, and the driving effect of all kinds of spatial influence factors on land use in Chongzhou is significant. The constructed Bayesian networks can capture the uncertainty of spatial influencing factors, especially in the case that all land use information cannot be captured at present, which highlights its advantages in applicability. In addition, fuzzy mathematical planning can supplement the role of land use planning on land use to a certain extent, quantify the impact including economy, environment and policy, help to understand the development of land use in an all-round way and provide some support for optimizing land use so as to promote the orderly, controllable and diversified development of cities. Based on the results of this study, this paper will provide some thoughts for urban land use from the following aspects: Firstly, based on the concept

of sustainable development, more consideration should be given to the environmental benefits and carbon emissions of land use under uncertain and flexible conditions, so as to formulate more promising land plans for cities. Second, based on the influencing factors of land use, planning should correctly grasp the law of land use development, strictly protect high-quality land and guard against the expansion of inefficient use of construction land. The third aspect is to rationally plan functional areas according to different geographical locations of urban space, give full play to regional resource endowments and advantages and formulate more targeted land schemes.

Due to the limitation of current technology and foundations, there is still room for improvement and optimization in this study. The goal of this study is to provide support for the decision-making system, which can reflect the information and elements needed in the decision-making process to a certain extent. The Bayesian network and fuzzy mathematical programming method can capture more uncertainty, which is verified to be correct to in some sense, but the construction of the model is limited to the primary stage and cannot reflect all information and components. The real system is often much more complex than the idea, so it needs to be further explored to clarify more details inside the model, which have been rarely probed so far. The current challenges also include the quantification of external influencing factors, the accessibility of spatial data and the reasonable setting of fuzzy parameters. The uncertainty of land use will continue to exist in the future. Although, considering specific problems, the choice of models and methods may be different, it is still necessary to ensure that models and methods do not have a dominant impact on the research results, and more attention should be paid to practical problems.

## 6. Conclusions

From the perspective of land supply-side structural reform, this study took cultivated land and urban construction land in Chongzhou in 2017 as the main research objects, carried out a two-stage study on land use optimization and drew the following conclusions. The spatial distribution of joint impact probability of cultivated land and urban construction land shows that the best land space of cultivated land is in the middle and south, while the regional conversion probability near the east is low, which may be caused by the distribution of water sources and rainfall. The joint impact probability of urban construction land is relatively smooth. Except for the probability of the western reserve being low, the joint impact probability of other parts is generally similar. Compared with other areas, this area is far from the water area, so the overall probability is slightly affected. From the perspective of spatial performance, the form of urban construction land is relatively concentrated, while cultivated land is located in the periphery of urban construction land. In the long run, urban construction land has the dynamic characteristics of "catching up" with cultivated land. If the scale of urban construction land is not controlled and high-quality cultivated land resources are not protected, there will be a risk of imbalance in urban spatial structure, and the space of forest land, irrigated grassland and protected areas will be more tense. In addition to the reclamation of wasteland, Chongzhou can also make greater efforts in land consolidation and reclamation to make full use of high-quality agricultural resource conditions. Construction land can be appropriately increased, but it should not exceed the scale control of construction land. The carbon emissions from production, construction and living in Chongzhou can be fully absorbed by the local carbon sink, which can be self-sufficient, and the carbon emissions are well controlled. Therefore, Chongzhou plays an important role as an ecological conservation area in Chengdu. The results show that the increase in construction land can occur in the central and western parts of Chongzhou, which may be caused by the radiation force of various conditions inside Chongzhou and the external radiation force of the core area of Chengdu. The Monte Carlo verification results show that when the conversion probability is more than 50%, the cultivated land can be turned into urban construction land, with an accuracy of 91.99%, passing the test. In addition, the study provides a methodological reference for the formulation of more sustainable and ecologically friendly land schemes, explores the

land use impact mechanism and land use impact under uncertain conditions, expands the perspective and method of land use impact mechanism research, further deepens the understanding of uncertainty in land use process and provides certain theoretical significance for land use development research.

**Author Contributions:** J.Y. and B.Q. conceived and designed the research; A.D. and B.Q. collected, managed, A.D. verified the data; M.Z. and S.L. calculated and analyzed the data and the results; J.Y. and M.Z. wrote the manuscript. All authors have read and agreed to the published version of the manuscript.

**Funding:** This research was supported by National Natural Science Foundation of China No. 41401631.

**Institutional Review Board Statement:** Not applicable.

**Informed Consent Statement:** Not applicable.

**Data Availability Statement:** Not applicable.

**Conflicts of Interest:** The authors declare that they have no conflict of interest.

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
