# Peer review of "A Two-Stage Fuzzy Optimization Model for Urban Land Use: A Case Study of Chongzhou City"

_sustainability, doi:10.3390/su132413961_

Round 1
Reviewer 1 Report
1. The Abstract is poorly written due to poor logic of storytelling and improper background for research question.
2. The language of this manuscript should be improved due to a poor readability.
3. Introduction is poorly written due to poor logic of knowledge gap summarizing and research question putting forward.
4. Research objectives were presented in an unclear way.
5. Due to too simplicity, Figure 2 is not necessary which can be deleted.
6. It is suggested to summarize out those spatial factors considered in Section 2.1.1 (lines 204-06) through similar studies via citations.
7. Functions of water bodies, such as rivers, lakes and seas (line 224) are not the same for a certain city; although are the same in an abstract sense. Statements like these should not be seen when explaining concrete situation of the study case.
8. It is improper to introduce basic information of the study case in Section 3 and to explain methods in Section 2. Because sentences like based on the conditions of the study case have been mentioned many times before being introduced. Poor logic.
9. Legend of Figure 6, 10 and 11 are too small to be readable.
10. An elevation map of the study area is needed.
11. The pixel size of raster data is 250m×250m (line 462) is improper to analyze an area with over one third total size is high mountains (lines 391-2).
12. Figures such as Figure 15 should be independent when being read.
13. It is improper to explain Monte Carlo method in Section 4.5. It is unusual writing format.
14. The three thoughts provided via results are already known. Indeed, thoughts such as intensive use of the current construction land cannot be drawn directly from results.
Author Response
RESPONSES TO REVIEWER #1’ COMMENTS
We are very grateful to the reviewer for his/her insightful review. The comments and suggestions provided have contributed a great deal to improving the manuscript. According to these, we have made efforts in revising the manuscript, with the details explained as follows:
Note: For the reviewer’s convenience in re-reviewing, we have divided his/her comments into several parts with ordinal numbers, and responded to them on a point-by-point basis.
COMMENT 1 –1. The Abstract is poorly written due to poor logic of storytelling and improper background for research question.
Response: We are thankful for the reviewer’s valuable suggestion, and have completed a thorough proof-reading. You can see all the revisions in the revised version which has been uploaded to the submission system. We used a revision model of the word, where the revisions can be seen more clearly.
COMMENT 2. The language of this manuscript should be improved due to a poor readability.
Response: We are thankful for the reviewer’s valuable suggestion, and have completed a thorough proof-reading. You can see all the revisions in the revised version which has been uploaded to the submission system. We used a revision model of the word, where the revisions can be seen more clearly.
COMMENT 3. Introduction is poorly written due to poor logic of knowledge gap summarizing and research question putting forward.
Response: We are thankful for the reviewer’s valuable suggestion, and have completed a thorough proof-reading. You can see all the revisions in the revised version which has been uploaded to the submission system. We used a revision model of the word, where the revisions can be seen more clearly.
COMMENT 4. Research objectives were presented in an unclear way.
Response: Thank you for your suggestion. We review the objectives of this study and rewrite the following : ' Therefore, the purpose of this study is to analyze the impact mechanism of land use and optimize the structure of land use under uncertain conditions, in order to expand the perspective and method of land use impact mechanism research. ' See the revised edition in lines 145-147.
COMMENT 5. Due to too simplicity, Figure 2 is not necessary which can be deleted.
Response: Thank you for your comments. We have deleted Figure 2.
COMMENT 6. It is suggested to summarize out those spatial factors considered in Section 2.1.1 (lines 204-06) through similar studies via citations.
Response: Thank you for your suggestion. Based on the review of related research, we have summarized the spatial factors. Please see in the revised edition line 209-212.
COMMENT 7. Functions of water bodies, such as rivers, lakes and seas (line 224) are not the same for a certain city; although are the same in an abstract sense. Statements like these should not be seen when explaining concrete situation of the study case.
Response: Thank you for your comments. Although water has universal and general significance for agricultural development and urban construction, rivers, lakes and oceans have different roles in different cities. We have deleted the original description of the water and have r written it as follows: “For agriculture, water supply will directly have an important impact on crop quality and yield. Rice, the main food crop in Chongzhou City, will affect its yield when the lack of water resources. At the same time, water is an important resource guarantee for urban construction. Sufficient water supply will support urban life and production, and vice versa will limit urban sustainable development. In order to improve the quality of water resources, provide a good resource endowment for urban life and production, and improve the ecological and landscape conditions of the city, Chongzhou City has implemented measures such as ' water control ten ', river ( lake ) chief system, black and odorous water treatment and comprehensive water environment management. Therefore, this shows that the current agricultural development and urban construction are both limited by water and dependent on water. Based on this, in the revised version, we supplement the role of water in the study area of the article to specify the role of water sources in Chongzhou ' s agricultural development and urban construction.” Please view the results in article in lines 235-249.
COMMENT 8. It is improper to introduce basic information of the study case in Section 3 and to explain methods in Section 2. Because sentences like based on the conditions of the study case have been mentioned many times before being introduced. Poor logic.
Response: Thank you for your valuable comments on the article.We delete basic information of the study case in Section 2 and supplement some information in Section 3 to make the article more logical.
COMMENT 9. Legend of Figure 6, 10 and 11 are too small to be readable.
Response: Thank you for your comments.We have replaced the blurry figure 6 with a slightly sharper one,enlarged figures 10 and 11. It may still affect the view, but we have to adjust it this way to make the format beautiful. Otherwise,we are sending you the picture file for your review.
COMMENT 10. The pixel size of raster data is 250m×250m (line 462) is improper to analyze an area with over one third total size is high mountains (lines 391-2).
Response: Thank you for your comments.The reason why we choose the pixel size of raster data is 250m×250m, because the data that can be collected is limited. Such pixel size is more conducive to the analysis of this paper with Netica.
COMMENT 11. Figures such as Figure 15 should be independent when being read.
Response: Thank you for your comments. We have enlarged Figure 15, which originally stands independently.
COMMENT 12. It is improper to explain Monte Carlo method in Section 4.5. It is unusual writing format.
Response: Thank you for your opinion. We make the following modifications to this part to make it more scientific and readable: “It is a method that can be systematically and comprehensively explored, which is superior to the parameters of isolation, effectiveness analysis and single-class model in performance, especially in the ability to show the coupling error caused by coupling through Monte Carlo test. Therefore, this study uses Monte Carlo thought to verify the transformation rules, and completes simulation and calculation with ArcGIS and Excel.” Please see in the revised edition in lines 731-736.
COMMENT 13. The three thoughts provided via results are already known. Indeed, thoughts such as intensive use of the current construction land cannot be drawn directly from results.
Response: Thank you for your opinion. We have revised and expanded the conclusions of this paper’ s policy recommendations based on the specific circumstances and research results of case studies. Please see in the revision in lines 796-804.
Generally, we very much appreciate the reviewers’ reviews. The provided comments/suggestions have contributed greatly to improving the manuscript.

Reviewer 2 Report
General comments:
This study analyzed the influencing mechanism of land use and the optimization of land use structure in two stages using Bayesian network and fuzzy mathematical programming. This study is useful, but several problems should be addressed.
Specific comments:
- The authors should add more relevant references to support the views in the introduction section, while shorting the introduction to make it focused on this research issue. The knowledge gaps should be analyzed in-depth in the introduction section to highlight the significance of this study. The selection basis for the study area and the study period should be briefly introduced in the last paragraph.
- The methodology section should be reorganized to make it more readable, and some common-sense introductions need to be deleted. The flowchart should be modified to make it self-explanatory.
- The results and discussion section should be reorganized, and moved the descriptions of data, parameter settings, and method to other sections. In lines 469-482, the explanation of terms should be placed in the figure title. The subfigures should be explained in the figure title, e.g., figures 10-12, 15. The authors should modify all the figures to make the text clear.
- The author should add a new section (i.e., discussion) to clarify the comparison with related research methods, and the advantages, disadvantages and future research topics of this study. Please remove all the explanations and references in results part to the discussion section.
- The language needs major revision.
Author Response
RESPONSES TO REVIEWER #2’ COMMENTS
We are very grateful to the reviewer for his/her insightful review. The comments and suggestions provided have contributed a great deal to improving the manuscript. According to these, we have made efforts in revising the manuscript, with the details explained as follows:
Note: For the reviewer’s convenience in re-reviewing, we have divided his/her comments into several parts with ordinal numbers, and responded to them on a point-by-point basis.
COMMENT 1. The authors should add more relevant references to support the views in the introduction section, while shorting the introduction to make it focused on this research issue. The knowledge gaps should be analyzed in-depth in the introduction section to highlight the significance of this study. The selection basis for the study area and the study period should be briefly introduced in the last paragraph.
Response: Thank you for your comments. We have significantly revised the introduction section.We short the introduction to make it focused on this research issue, and introduce selection basis for the study area and the study period, which can be seen in lines 144-152.
COMMENT 2. The methodology section should be reorganized to make it more readable, and some common-sense introductions need to be deleted. The flowchart should be modified to make it self-explanatory.
Response: Thank you for your opinion. The methodology section has been modified, and we have reorganized and deleted some common sense concepts and modified the flowchart. In addition, to ensure the overall readability of the article, we chose to retain some conceptual principles. Please see in the second part of the article.
COMMENT 3. The results and discussion section should be reorganized, and moved the descriptions of data, parameter settings, and method to other sections. In lines 469-482, the explanation of terms should be placed in the figure title. The subfigures should be explained in the figure title, e.g., figures 10-12, 15. The authors should modify all the figures to make the text clear.
Response: Thank you for your comments.We have added the explanation of terms in the title of figure 8 and figure 9.In addition,we have also redefined the NaturalEndowment as follows:"NaturalEndowment" refers to natural abilities or qualities.And keep the capitals in the text and charts consistent.Besides this,some synonyms have been replaced.Finally, we rearranged the numbers of the charts and their corresponding positions in the text.
COMMENT 4. The author should add a new section (i.e., discussion) to clarify the comparison with related research methods, and the advantages, disadvantages and future research topics of this study. Please remove all the explanations and references in results part to the discussion section.
Response: Thank you for your comments. We have significantly revised the original version.We add a discussion section to explain our results and clarify the comparison with related research methods.We also remove “4.6. Policy Suggestions” to this part. Finally, we propose the uncertainty of land use will continue to exist in the future,so it needs to be further explored .
COMMENT 5. The language needs major revision.
Response: We are thankful for the reviewer’s valuable suggestion, and have completed a thorough proof-reading. You can see all the revisions in the revised version which has been uploaded to the submission system. We used a revision model of the word, where the revisions can be seen more clearly.
Generally, we very much appreciate the reviewers’ reviews. The provided comments/suggestions have contributed greatly to improving the manuscript.

Reviewer 3 Report
The topic is current and important, but the examination of the issue is unnecessarily complicated. Right at the beginning: the abstract is too long (more than 330 words). It must be clarified and shortened (max 200 words). The starting points are not concrete enough. The authors must pay attention to the main theoretical aspects and connect them to the results when deepening the conclusions. Important concepts such as “new-type urbanization" cannot be left without explanation and references. The methodology is also self-serving and does not help to solve the real problem. It must be emphasized how the Monte Carlo method and Bayesian neural network actually help landscape planning and rural development. The usable results must be placed in an international context. Mathematical analysis is very good and abundant but should be supported by qualitative studies (empirical surveys). A number of small errors can be found: what are the sources of Figures? Overall, the article is characterized by excellent mathematical reasoning, but there is no real discussion, conclusion, a well-defined message.
Author Response
RESPONSES TO REVIEWER #3’ COMMENTS
We are very grateful to the reviewer for his/her insightful review. The comments and suggestions provided have contributed a great deal to improving the manuscript. According to these, we have made efforts in revising the manuscript, with the details explained as follows:
Note: For the reviewer’s convenience in re-reviewing, we have divided his/her comments into several parts with ordinal numbers, and responded to them on a point-by-point basis.
COMMENT 1. Right at the beginning: the abstract is too long (more than 330 words). It must be clarified and shortened (max 200 words).
Response: We are thankful for the reviewer’s valuable suggestion, and have completed a thorough proof-reading. You can see all the revisions in the revised version which has been uploaded to the submission system. We used a revision model of the word, where the revisions can be seen more clearly.
COMMENT 2. The starting points are not concrete enough. The authors must pay attention to the main theoretical aspects and connect them to the results when deepening the conclusions.
Response: We are thankful for the reviewer’s valuable suggestion, and have completed a thorough proof-reading. You can see all the revisions in the revised version which has been uploaded to the submission system. We used a revision model of the word, where the revisions can be seen more clearly.
COMMENT 3. Important concepts such as “new-type urbanization" cannot be left without explanation and references. The methodology is also self-serving and does not help to solve the real problem. It must be emphasized how the Monte Carlo method and Bayesian neural network actually help landscape planning and rural development. The usable results must be placed in an international context.
Response: Thank you for your opinion. We supplement the concept of new urbanization as follows. The new urbanization strategy proposed by the Chinese government is people-oriented urbanization, focusing on the coordination of population, economy, society and ecological environment. Compared with the traditional urbanization, the new urbanization emphasizes that the urbanization construction pays more attention to the level and quality of urban development, especially the optimization of urban spatial structure layout, and emphasizes the coordinated development of economic construction and ecological civilization. Please see in the revised edition in lines 157-159.
For the second question, according to the suggestions of other second commentators, we have revised the common sense concept and structure of the methodological part. Please see in the revised edition in Section 2.
In addition, bayesian network is based on the complex occurrence probability reasoning between variable factors, which can be used to evaluate the correlation between complex variable factors and the probability of occurrence between events. The influencing factors of land use cover roads, water sources, slope and other aspects, with many influencing factors and complex connections. Bayesian network with its good graphical description method clearly shows the complex relationship between land use change and its driving factors and its change process, which is conducive to the in-depth study of the mechanism of land change. Monte Carlo method is a probability-based calculation method for uncertain factors, which is applied to the field relationship operation of cellular automata to determine the conversion state of land use in each unit. This deepens the understanding of uncertain factors in land use system, helps to study complex, uncertain and dynamic land use system and its influencing mechanism, and obtains more objective and scientific research results.In this paper, we add an explanation of the role of Monte Carlo and Bayesian neural network methods in analyzing land use impact mechanism and optimizing land use structure. Please see in the revised edition in lines 186-188, 724-726.
Moreover, in the discussion of the results, the research significance and influence are expanded based on the international background. Please see in the revised edition. Please see in the revised edition in lines 841-846.
Reference
Boqiang Lin, Junpeng Zhu, Impact of China's new-type urbanization on energy intensity: A city-level analysis. Energy Economics 2021, 99-105292
COMMENT 4. Mathematical analysis is very good and abundant but should be supported by qualitative studies (empirical surveys). A number of small errors can be found: what are the sources of Figures? Overall, the article is characterized by excellent mathematical reasoning, but there is no real discussion, conclusion, a well-defined message.
Response: Thank you for your comments. The main sources of data are mentioned in the introduction of several models. For example, the Outline of Food and Nutrition Development in China (2014-2020),the Chongzhou City General Planning (2016-2035),the General Plan of Chengdu City (2016-2035),the Chengdu Yearbook 2018, China Energy Statistical Yearbook 2018,Chongzhou Yearbook 2018 and Sichuan Statistical Yearbook 2018.Elevation and spatial land use data from the geographical science and resources institute of the Chinese Academy of Sciences and Tsinghua University. Spatial rainfall data were collected from National Tibetan Plateau Scientific Data Center, and spatial data such as roads, water systems, attractions and commercial centers are derived from OpenStreetMap database.
Generally, we very much appreciate the reviewers’ reviews. The provided comments/suggestions have contributed greatly to improving the manuscript.

Round 2
Reviewer 1 Report
In attachment

Author Response
RESPONSES TO REVIEWER #1’ COMMENTS
Thank you very much for giving us an opportunity to revise our manuscript. We sincerely appreciate the editor and reviewers for their constructive comments and suggestions on our manuscript. We have carefully studied the reviewers’ comments and detailed suggestions. According to these suggestions, we have carefully revised the original manuscript. All revised portions are marked in red in the resubmitted manuscript that we submit for your kind consideration. Additionally, the language in the manuscript has been checked by a professional English editing service before resubmission to this journal.
The following issues should be addressed in the paper:
- In Abstract, it cannot found research question(s) or/and knowledge gap(s), and the reason(s) why experiment has been designed via using the adopted methods. Furthermore, the mentioned "territory spatial planning", actually, has no direct relations with the background mentioned.
Response: Thank you for your comments. We found research question and the reason why experiment has been designed via using the adopted methods. Furthermore, land use structure optimization and spatial optimization together provide an important theoretical basis for the formulation of territorial spatial planning. Because the uncertainty of spatial factors will affect land use decision-making, and incorporating uncertainty into the optimization of land use structure has gradually become one of the important modeling directions in the future.
- Background information, i.e., the first paragraph of the Introduction, lacks sufficient support especially judged from the perspective of literature review.
Response: Thank you for your comments. We have added some references to the relevant arguments in the introduction and have removed inappropriate or cumbersome statements.
- It is so strange that one paragraph introducing part information of the study area inserted in the Introduction without any logic linkage with its former paragraph.
Response: Thank you for your comments. We have removed the paragraph that introduces the research area, and we apologize for the inconvenience of such a problem in the previous revision.
- The mentioned potential innovation has nothing to do with background information as well as the possible knowledge gap summarized by authors.
Response: Thank you for your comments. We talked about it in the modified version. “In view of the shortcomings of the above models, the purpose of this paper is to fully consider the uncertain factors in the land use system, optimize the land use quantity and the spatial layout by combining cellular automata model, so as to provide quantitative support for decision makers.”
Please see it at L.156-159.
- Words and phrases of Figure 1 are too small.
Response: Thank you for your comments. We enlarge words and phrases of Figure 1.
- A location map is needed.
Response: Thank you for your comments. We add a location map .
- Many comments provided by the reviewer last time have not been accepted and revised; while a responding letter lacks which means insufficient support for rejecting the mentioned comments have been provided by authors.
Response: Thank you for your comments. We apologize for the inadequate response last time, and we have made article revisions in response to the last reviewer's comments. In particular, we have added references to the relevant arguments in the introduction section and added a map of the location of the study area.

Reviewer 2 Report
All my comments have been well revised.
Author Response
Author's Reply to the Review Report (Reviewer 2)
We appreciate you for your valuable advice, which is of great significance for us to improve our paper.

Reviewer 3 Report
The study was improved by the fixes. In the future, it should also be noted that the international readership does not always understand the topic. That is why everything needs to be made clear.
Author Response
Author's Reply to the Review Report (Reviewer 3)
We appreciate you for your valuable advice, which is of great significance for us to improve our paper.
Comments: The study was improved by the fixes. In the future, it should also be noted that the international readership does not always understand the topic. That is why everything needs to be made clear.
Response: According to your suggestion, we explained this.
Please see it at L.843-848.